# Explainable Artificial Intelligence (XAI) for generation of library of novel potential acetylcholinesterase inhibitors

**Nirajan Bhattarai, Marvin K Schulte***

**Department of Biomedical and Pharmaceutical Sciences, Idaho State University, Pocatello,ID**

Data-driven deep generative models hold promise in generalized inverse design but face limitations due to their requirement for extensive datasets.  Traditional machine learning methods primarily predict desired compounds within existing compound spaces rather than generating entirely novel compounds (Bilodeau et al., 2022; Lu et al., 2022). To address this issue, we purpose Local Interpretable Model-agnostic Explanations (LIME) within Explainable Artificial Intelligence (XAI) to map crucial structural features for desired pharmacological effects, and generate novel acetylcholinesterase inhibitors from the learned structural features.

We developed a predictive model using TPOT automated machine learning with circular fingerprint features on the ChEMBL220 dataset of acetylcholinesterase inhibitors. Subsequently, we identified highly influential fragments from known acetylcholinesterase inhibitors predicted by the model. Fragment expansion to new acetylcholinesterase analogs was facilitated by Pat Walter's method (Walters, P. (2020), leveraging the CReM Python library for fragment expansion and the REOS Python class for smart pattern-based filtering. Novel acetylcholinesterase inhibitors were selected from the generated compounds through screening by Quantitative Estimate of Druglikeness (QED) and Synthetic Accessibility Score (SAS).

The optimal AutoML pipeline was found to be GradientBoostingClassifier with a cross-validation score (5 folds) of 0.85 +/- 0.015 (95% CI = 0.029). Across various species including human test, human independent, eel, mouse, cow, ray, and mosquito, the (AUC ROC, accuracy) scores were (0.92, 0.86), (0.70, 0.74), (0.76, 0.75), (0.89, 0.83), (0.87, 0.81), (0.74, 0.66), and (0.71, 0.81) respectively. After analyzing the human test and human independent sets, a prediction probability threshold of 0.8 was identified as optimal for accurate prediction of the positive class.Using LIME interpretation on the test dataset, we identified the ten most important features and 356 key fragments. Expansion of these fragments resulted in a total of 136,156 generated compounds. Further refinement included selecting compounds with a quantitative estimate of drug likeness (QED) cutoff of 0.7 and synthetic accessibility scoring (SAS) cutoff of 6, yielding 66,387 compounds. From these, 1,506, 464, 116, 23, and 6 compounds were identified as potential novel acetylcholinesterase inhibitors with prediction probability thresholds of 0.5, 0.6, 0.7, 0.8, and 0.9, respectively, based on screening through predictive modeling.

Integration of Explainable AI (XAI) techniques in predictive modeling of acetylcholinesterase inhibitors shows significant potential in advancing drug discovery. By combining TPOT AutoML with circular fingerprint features and leveraging the ChEMBL220 dataset, we developed a high-performing model with robust cross-species predictive accuracy. Utilizing LIME for feature interpretation enabled the identification of critical features and structural fragments, facilitating the systematic generation of a diverse library of potential inhibitors. This approach not only enhances model interpretability but also supports the discovery of novel drug candidates, underscoring the transformative role of XAI in drug development.

**Keywords:** Explainable AI (XAI), Predictive modeling, Acetylcholinesterase inhibitors, Drug discovery, Novel compound generation