# OpenReview forum: "Explainable Artificial Intelligence (XAI) for generation of library of novel potential acetylcholinesterase inhibitors"
_IEEE.org/ICIST/2024/Conference — IEEE ICIST 2024 Conference Submission_

### Official Review · Reviewer_YSwE · 2024-08-21
**Explainable Artificial Intelligence (XAI) for generation of library of novel potential acetylcholinesterase inhibitors**

**Rating:** 3
**Confidence:** 3

**Review:**

This paper investigates the application of explorable artificial intelligence in drug development. However, the paper suffers from the problem of limited contribution, and I cannot recommend its publication.

---

### Official Review · Reviewer_MdPh · 2024-08-21
**Accept**

**Rating:** 7
**Confidence:** 3

**Review:**

This paper primarily investigates the application of Explainable Artificial Intelligence in drug research. Using the ChEMBL220 dataset, a powerful cross-species predictive model was developed that can identify key features. This is a highly meaningful study, and I recommend it for publication.

---

### Official Review · Reviewer_teev · 2024-08-22
**High-performing model, novel, publication**

**Rating:** 7
**Confidence:** 3

**Review:**

This paper develops a high-performing model and identifies highly influential fragments. By integrating Explainable Artificial Intelligence (XAI) techniques in the predictive modeling of acetylcholinesterase inhibitors and using LIME explanations, multiple important features and key fragments were identified, and ultimately, multiple potential novel acetylcholinesterase inhibitors. This paper presents a novel and informative approach and is recommended for publication.

---

### Decision · Program_Chairs · 2024-09-08

Accept (Oral)